# Optimization of Dynamic SSVEP Paradigms for Practical Application: Low-Fatigue Design with Coordinated Trajectory and Speed Modulation and Gaming Validation

**DOI:** 10.3390/s25154727

**Published:** 2025-07-31

**Authors:** Yan Huang, Lei Cao, Yongru Chen, Ting Wang

**Affiliations:** School of Information Engineering, Shanghai Maritime University, Shanghai 201306, China; athrunhy12138@gmail.com (Y.H.); 202330310297@stu.shmtu.edu.cn (Y.C.); 202430310253@stu.shmtu.edu.cn (T.W.)

**Keywords:** SSVEP, BCI, dynamic stimulation paradigm, speed modulation, trajectory modulation, fruit-slicing game

## Abstract

Steady-state visual evoked potential (SSVEP) paradigms are widely used in brain–computer interface (BCI) systems due to their reliability and fast response. However, traditional static stimuli may reduce user comfort and engagement during prolonged use. This study proposes a dynamic stimulation paradigm combining periodic motion trajectories with speed control. Using four frequencies (6, 8.57, 10, 12 Hz) and three waveform patterns (sinusoidal, square, sawtooth), speed was modulated at 1/5, 1/10, and 1/20 of each frequency’s base rate. An offline experiment with 17 subjects showed that the low-speed sinusoidal and sawtooth trajectories matched the static accuracy (85.84% and 83.82%) while reducing cognitive workload by 22%. An online experiment with 12 subjects participating in a fruit-slicing game confirmed its practicality, achieving recognition accuracies above 82% and a System Usability Scale score of 75.96. These results indicate that coordinated trajectory and speed modulation preserves SSVEP signal quality and enhances user experience, offering a promising approach for fatigue-resistant, user-friendly BCI application.

## 1. Introduction

Brain–computer interface (BCI) technology enables direct control of external devices by decoding neural signals, bypassing traditional input methods and conventional motor pathways. This approach facilitates communication and control without relying on physical movement or peripheral nervous system mediation [1,2,3,4]. This distinctive human–machine interaction modality has been extensively applied across multiple domains, particularly in clinical settings such as medical rehabilitation assistance [5], education [6,7], security [8], and entertainment gaming [9,10]. Electroencephalography (EEG) persists as the optimal selection for BCI implementation among neurosensing technologies, which can be attributed to its non-invasive nature, superior temporal resolution, practical deployability, spatial adaptability, and operator convenience [11,12]. Steady-state visual evoked potential (SSVEP)-based BCI systems [13,14], which utilize SSVEP as control signals, are widely favored because they require little user training [15] and provide a high SNR [16,17].

Nevertheless, conventional SSVEP-BCIs predominantly employ spatially fixed visual stimuli, which may induce ocular strain, cephalalgia, and attentional decline during prolonged use. These adverse effects compromise SSVEP signal quality and consequently degrade BCI system performance [18]. Recent studies suggest that dynamic visual stimuli (e.g., moving targets) may improve user comfort and attention engagement compared to spatially fixed flickers. However, the neurophysiological effects of motion parameters—primarily trajectory and speed—remain insufficiently understood due to methodological heterogeneity and limited exploration of complex motion patterns. Pitchaimmuthu et al. [19] investigated higher-order visual cortical processing of motion information by superimposing two horizontal periodic motion stimuli (2.1 Hz and 2.4 Hz) onto a 6.1 Hz luminance-flickering target. Notably, congenital cataract patients exhibited no detectable intermodulation frequency components compared to healthy controls, indicating impaired processing of combined luminance–motion visual stimuli. Punsawad et al. [20] developed a novel paradigm employing laterally alternating flickering bars to create motion illusions, achieving approximately 80% classification accuracy for left–right face recognition while mitigating ocular fatigue. Kanoga et al. [21] explored the impact of head-tracking movements on system performance, demonstrating performance degradation with increasing head movement speed, particularly under vertical displacement conditions. These findings emphasize the necessity of evaluating motion-induced interference when integrating positional dynamics into traditional paradigms. Duan et al. [22] systematically investigated a 3 × 3 moving visual stimulus matrix through vertical motion experiments with four velocities and phase intervals, identifying optimal parameters achieving comparable performance to static paradigms. Zhang et al. [23] implemented an SSVEP-BCI using 12 flicker frequencies (6.2–16.6 Hz, 0.9 Hz intervals) to encode randomly directed motion stimuli, reporting 86.67% average recognition accuracy versus 89.26% for static conditions. Li et al. [24] proposed a dual-modulation paradigm combining luminance variation with horizontal motion (0–0.6 Hz) and their experimental results demonstrated that 0.2 Hz movement cycles yielded superior visual perception quality. In summary, while dynamic stimuli have demonstrated promise in enhancing SSVEP-BCI usability, the current research is still fragmented and lacks systematic investigation into how motion dynamics shape neural responses.

Furthermore, contemporary research has explored the integration of gaming elements with SSVEP-BCI systems to enhance both performance and user comfort. Parafita et al. [25] developed an SSVEP-BCI gaming application for clinical trials, featuring two-frequency stimuli to control spacecraft lateral movements for obstacle avoidance. Experimental results indicated that participants maintained over 95% mean accuracy during successful gameplay completion. Cruz et al. [26] presented “Kessel Run,” a cooperative multi-user gaming platform requiring synchronized interactions where dual users jointly controlled a spacecraft’s thrusters via SSVEP signals to evade projectiles. The cooperative gameplay required obstacle-free navigation to be maintained for 2 min, with participants reporting strong interactive and collaborative experiences. Z-Valero et al. [27] implemented an SSVEP-BCI-controlled game utilizing a 15 Hz stimulus presented as a circular checkerboard pattern moving from the screen periphery to the center. Users were tasked with concentrating (with attention levels monitored through SSVEP power spectral density and EEG background noise analysis) to halt its movement, achieving an average accuracy of 84%. Experimental results demonstrated the game’s efficacy in training visual tracking and sustained attention capabilities. Existing studies indicate that, while visual stimuli typically function as isolated control signals in current implementations, their deeper integration with gaming mechanics remains underdeveloped. This underscores the significance of investigating SSVEP-BCI systems that harmoniously combine dynamic visual stimuli with immersive gaming paradigms.

Building upon previous studies, this work addresses a key limitation in the current dynamic SSVEP-BCI research—namely, the lack of systematic coordination between stimulus trajectory and motion speed. Although prior research has demonstrated the feasibility of introducing motion trajectories or gaming elements, most existing paradigms adopt either fixed speed settings or single trajectory forms, without fully evaluating how their combinations influence both signal quality and user experience. To bridge this gap, we designed a set of visual stimulation conditions that incorporate four flicker frequencies (6 Hz, 8.57 Hz, 10 Hz, and 12 Hz), three motion speed ratios (1/20, 1/10, and 1/5 of the flicker frequency), and four trajectory types (static, sinusoidal, square, and sawtooth). An offline experiment was conducted to systematically assess how different trajectory–speed combinations affect SSVEP responses and subjective cognitive load. In addition, an online fruit-slicing game was developed as a practical BCI application to validate the real-world feasibility of the proposed stimulation design. Subjective feedback on comfort and usability was also collected, highlighting the potential of this approach to improve the immersion and user-friendliness of SSVEP-based BCI systems.

## 2. Materials and Methods

### 2.1. Experimental Configuration

We employed a 23.8-inch LCD (DELL S2417DG, Dell Inc., Round Rock, TX, USA) featuring a 2560 × 1440 pixel resolution and 60 Hz refresh frequency for visual stimulus delivery. The trial protocols were developed using web technologies including HTML, JavaScript, and CSS. EEG data were captured through a 16-electrode Econtek iRecorde (Shanghai Niantong Intelligence & Technology Co., Ltd., Shanghai, China) system operating at 500 samples per second. Reference electrodes were strategically located at standardized GND and REF positions. Following the 10–20 international electrode placement standard, seven strategically chosen recording sites (Pz, POz, PO3, PO4, Oz, O1, O2) served both real-time monitoring and post-experiment analysis purposes. Signal conditioning involved dual-stage processing: initial band-pass filtration (0.15–200 Hz) followed by 50 Hz notch filtration to mitigate electrical interference.

### 2.2. Subjects

Two groups of subjects took part in the study, including 25 people. The offline experiment involved 17 individuals (12 males and 5 females) aged between 21 and 26 years (average age 24), while 12 individuals (10 males and 2 females), aged 18–26 with an average age of 22, were involved in the online session. Four of them also joined the offline experiment and online experiment. All subjects were healthy individuals with normal or corrected-to-normal vision and no history of neurological or psychiatric disorders. Before the experiment, subjects were guided through the experimental procedure and explicitly informed of their right to withdraw at any time. In addition, they signed an informed consent form.

### 2.3. Experimental Design

#### 2.3.1. Offline Paradigm Design

The experimental display presented in Figure 1 contains four 180 × 180 pixel circular stimuli positioned at each screen corner, operating at 6 Hz (top left), 8.57 Hz (top right), 10 Hz (bottom left), and 12 Hz (bottom right), respectively.

The flicker frequency was achieved using a sinusoidal sampling-based encoding method, while brightness modulation was implemented through a JFPM [28] approach. Specifically, the target brightness was controlled by adjusting a stimulation sequence s(f,φ,i) which corresponds to the stimulus frequency *f* and phase φ as shown in the formula below:(1)$$s\left(f,\varphi ,i\right)=\frac{1}{2}\times \left\{1+\sin\left[2\pi f(i/R)+\varphi \right]\right\}$$

Here, i denotes the index of the current frame in the stimulation timeline, and R specifies the display’s update frequency, fixed at 60 Hz in this experiment. The stimulus modulation function s(f,φ,i) produces luminance values normalized within the range [0, 1] where a value of 0 indicates complete darkness and 1 signifies full brightness. For all conditions in this study, the phase parameter φ was assigned a value of zero. The motion trajectories were defined as sinusoidal, square, and sawtooth waveforms (hereafter abbreviated as sin, squ, and saw, respectively). The instantaneous position P(F,i) of each stimulus target can be computed as follows:(2)$$P=A\cdot \sin\left[2\pi F(i/R)\right]$$(3)$$P=A\cdot \mathrm{sgn}\left\{\sin\left[2\pi F(i/R)\right]\right\}$$(4)$$P=2A\left\{f\left[(i/R)\right]-\left\lfloor f\left[(i/R)\right]+\frac{1}{2}\right\rfloor \right\}$$
where sgn(x) represents the sign function(5)$$\mathrm{sgn}\left(x\right)=\left\{\begin{array}{cc} 1, & \mathrm{if}\;x>0\\ 0, & \mathrm{if}\;x=0\\ -1, & \mathrm{if}\;x<0 \end{array}\right.$$

A represents the amplitude, and F represents the motion frequency. The motion frequencies are set to 1/5, 1/10, and 1/20 of the flicker frequency, which are abbreviated as 20, 10, and 5, respectively, in the following text. The motion trajectory of the flickering stimulus with an amplitude set to A = 180 is illustrated in Figure 2. Each stimulus moves along a predefined trajectory within its designated screen region, as shown in Figure 1a (located in the top-left, top-right, bottom-left, and bottom-right areas). Taking the center of the stimulus as the reference point, the trajectory length is 900 pixels, following a unidirectional, linear cyclic pattern. The movement is restricted to the local coordinate system of each stimulus container and does not span across the entire screen. Once the stimulus position exceeds the defined range, it resets to the initial position, forming a periodic motion cycle.

#### 2.3.2. Online Paradigm Design

As shown in Figure 3, to enhance the engagement and enjoyment of the experiment, this study incorporated the SSVEP-based visual stimulation task into an interactive fruit-slicing game in the online experiment. As illustrated in the figure, four fruit-shaped visual targets (lemon: 6 Hz, strawberry: 8.57 Hz, banana: 10 Hz, and apple: 12 Hz), identical to those used in the offline experiment, are positioned in four fixed regions on the screen. Each target integrates the trajectory and speed modulation strategies employed in the offline setting, exhibiting periodic motion and flickering at its designated frequency. These dynamic visual stimuli are designed to further enhance the strength and stability of the elicited SSVEP responses.

During the experiment, when a participant fixates on a specific fruit and successfully elicits the corresponding SSVEP signal, the system identifies the selected target and provides immediate “slicing-state” feedback. In this state, the selected fruit is virtually “sliced” on the screen, accompanied by a vivid slicing animation, as shown in the figure. This feedback mechanism enhances the user’s sense of immersion and operational achievement. Subjects are expected to accurately identify the goal fruit within a limited number of trials to complete the task.

#### 2.3.3. Validation of Flicker Frequency

To verify the accuracy and stability of the visual stimulus presentation frequency, we conducted systematic tests on the designed stimulation program to evaluate the discriminability and reliability of each flicker frequency used in the experiment. The stimulus targets underwent sinusoidal luminance modulation at preset frequencies, with a testing duration of five seconds. During this period, the timestamps of luminance peaks were recorded. Subsequently, the time intervals between adjacent peaks were calculated to estimate the instantaneous frequency of each cycle. The specific calculation formulas are as follows:(6)$$T_{i}=\frac{t_{i}-t_{i-1}}{1000}$$(7)$$f_{i}=\frac{1}{T_{i}}$$
where $t_{i}$ and $t_{i-1}$ represent the timestamps of two adjacent luminance peaks, $T_{i}$ is the duration of the i-th cycle, and $f_{i}$ is the instantaneous frequency of the i-th cycle. Next, the average frequency over all cycles was calculated as follows:(8)$$\bar{f}=\frac{1}{N}\sum_{i=1}^{N}f_{i}$$
where N is the total number of cycles and $\bar{f}$ is the average frequency of the entire stimulus presentation.

#### 2.3.4. Experimental Protocol

Before starting the offline and online experiments, we first conducted tests on the flickering stimuli to verify their stability. Only after confirming the stimulus stability did we proceed with the subsequent procedures.

By conducting the offline experiment, the proposed system was thoroughly evaluated and improved. It consists of 24 blocks incorporating four motion frequencies (static, 1/5, 1/10, and 1/20 of the flicker frequency) and three movement trajectories (sinusoidal wave, square wave, and sawtooth wave). Each frequency–trajectory condition was associated with 6 blocks. Each block was composed of four trials, each lasting for a duration of 6 s. It can be observed from Figure 1b that trials began with a 0.5 s central fixation cross cueing the target frequency, followed by simultaneous multi-target flickering for 5 s. After stimulation, all stimuli returned to their initial positions and ceased flickering for 0.5 s. Furthermore, all subjects completed the NASA-TLX questionnaire following the experiment.

The online experiment confirmed the practicality of the proposed approach. It tested three trajectories (static, sawtooth, and sinusoidal wave) and two velocities (0 and 1/20 flicker frequency). A horizontal motion condition (0.2 Hz) from Li et al. [24] was included for comparative analysis. Each condition comprised 20 blocks with four trials. Trials included 4 s of flickering stimuli followed by 0.5 s of visual feedback (fruit-slicing animation) upon successful recognition. Upon completion of the online experiment, the SUS questionnaire was administered to all subjects for usability assessment.

### 2.4. Target Classification Algorithm

Canonical Correlation Analysis (CCA) is a commonly applied method in signal processing that has been extensively applied in SSVEP decoding algorithms [29]. In this study, CCA was chosen as the decoding approach for both offline and online analysis of SSVEP. CCA operates through computation of canonical correlation coefficients linking recorded SSVEP signals with reference signals, with the goal of maximizing the linear correlation between the two variable sets.

One set of signals corresponds to the recorded EEG data $X=[x_{1},x_{2},\ldots ,x_{n}]$, while the other set Yi consists of template signals that match the frequencies used for visual stimulation which are specified as follows:(9)$$Y_{i}=\left(\begin{array}{c} \sin(2\pi f_{i}t)\\ \cos(2\pi f_{i}t)\\ \vdots \\ \sin(2\pi kf_{i}t)\\ \cos(2\pi kf_{i}t) \end{array}\right),\;t=\frac{1}{F_{S}},\;\frac{2}{F_{S}},\;\ldots ,\;\frac{N_{S}}{F_{S}}$$

Here, *i* denotes the index of stimulus targets, fi specifies the frequency, *k* indicates the harmonic count used in constructing the reference signals, and the total number of sampling points is specified as $N_{S}$. Given the brain’s low-pass filtering nature, which diminishes responses to high-frequency input, the harmonic number is limited to 3 in this study.

Linear combinations involving *X* and $Y_{i}$ are expressed as follows:(10)$$\begin{array}{cc} x & =X^{T}W_{X} \end{array}$$(11)$$\begin{array}{cc} y & =Y^{T}W_{Y} \end{array}$$
where $W_{X}$ and $W_{Y}$ are the respective matrices of weights. Based on this, for the i-th stimulus, the correlation with its reference signal is computed as follows:(12)$$\begin{array}{cc} \rho (f_{i}) & =\frac{E(x^{T}y_{i})}{\sqrt{E\left(x^{T}x\right)\;E\left(y^{T}y\right)}}\\ \; & =\frac{E(W_{X}^{T}XY_{i}^{T}W_{Y_{i}})}{\sqrt{E\left(W_{X}^{T}XX^{T}W_{X}\right)\;E\left(W_{Y_{i}}^{T}YY^{T}W_{Y_{i}}\right)}} \end{array}$$

The target $f_{s}$ frequency corresponding to the recognized stimulus can be determined when the total number of stimulus frequencies is denoted by *k* and is given by the following:(13)$$f_{s}=\max_{f_{i}}\rho \left(f_{i}\right),\;i=1,2,\ldots ,k$$

### 2.5. Performance Evaluation

The effectiveness of SSVEP-BCI systems is frequently measured using the Information Transfer Rate (ITR) [30], which functions as a core evaluation index, and its value can be calculated using a specific formula:(14)$$\mathrm{ITR}=\left[\log_{2}N+P\log_{2}P+\left(1-P\right)\log_{2}\left(\frac{1-P}{N-1}\right)\right]\times \frac{60}{T}$$

Here, N refers to the total number of selectable targets, P reflects the classification accuracy, and T denotes the length of the time window used for decoding a single command.

### 2.6. Subjective Assessment

#### 2.6.1. NASA-TLX

The NASA-TLX is a cognitive load evaluation scale [31] created by NASA, and it is among the most widely adopted subjective approaches for evaluating cognitive load. It is composed of six essential elements: mental demand, physical demand, time demand, performance, effort, and frustration. These are abbreviated respectively as MD, PD, TD, PE, EF, and FR. Initially, subjects are asked to conduct pairwise comparisons (a total of 15 pairs) to assess the relative importance of each dimension and assign weights, ensuring that the total sum of weights equals 15. Following this, each dimension is rated individually, typically on a scale from 0 to 100. The final weighted score is obtained by multiplying each dimension’s score by its assigned weight, summing the products, and dividing by 15, as represented by the following formula:(15)$$\mathrm{NASA\_TLX}\;=\frac{\sum_{i=1}^{6}w_{i}\cdot S_{i}}{\sum_{i=1}^{6}w_{i}}$$
where $S_{i}$ represents the rating for the *i*-th dimension (1–100), and $w_{i}$ denotes the corresponding weight derived from pairwise comparisons of the dimensions, ranging from 0 to 5.

#### 2.6.2. SUS

The SUS (System Usability Scale) is a concise and reliable standardized questionnaire. This approach was first proposed by John Brooke in the year 1986, created to rapidly evaluate users’ personal impressions regarding the usability of a product or system [32]. Subjects provide ratings reflecting their level of agreement, from complete disagreement to full agreement. The items are designed with alternating positive and negative wording to minimize response bias. After scoring adjustments, the scores are normalized to a 100-point scale, where a higher value reflects a greater level of perceived usability. The SUS is widely used in user experience evaluations across various systems due to its simplicity, reliability, and effectiveness.

For each odd-numbered item, one point is subtracted to obtain the adjusted score (if the original score is 5, it is recorded as 4), whereas, for even-numbered items, it is computed by reversing the original score on a 5-point scale (if the original score is 1, it is recorded as 4). The sum of the values from odd- and even-numbered items is then multiplied by 2.5 to produce a final score that ranges between 0 and 100, as indicated by the formula below:(16)$$G=2.5\times (S_{\mathrm{odd}}-S_{\mathrm{even}}+20)$$
where *G* represents the total score, and $S_{\mathrm{odd}}$ and $S_{\mathrm{even}}$ represent the odd-numbered and even-numbered questions, respectively.

The average score is approximately 68. A score above 70 indicates that the system performs well, while a score below 50 suggests a significant need for improvement.

## 3. Results

### 3.1. Flicker Frequency Verification Results

Table 1 presents the test results for flicker stimuli at four different frequencies. The test duration was 5 s for each condition, with the number of cycles N being 30, 42, 50, and 60, respectively. The average frequencies for the four groups were 5.9951 ± 0.0043 Hz, 8.5700 ± 0.0398 Hz, 10.0023 ± 0.0773 Hz, and 12.0013 ± 0.0432 Hz, indicating that the stimulus frequencies were both accurate and stable. Figure 4 illustrates the distribution of instantaneous frequencies under each frequency condition. It is evident that the instantaneous frequency fluctuations were minimal, with data points clustering tightly around their target frequencies, further confirming the precision and stability of the stimuli. Overall, these results demonstrate that the designed flicker stimulus program can reliably produce the preset frequencies, meeting the experimental requirements for visual stimulus frequency control.

### 3.2. Offline Experiment Results

According to previous research [33], the Oz channel is considered the optimal channel for observing SSVEPs. Located in the occipital region, Oz exhibits the strongest response to visual stimuli and clearly reflects the stimulus frequency components. Therefore, in this study, the SSVEP amplitude spectrum was calculated using the signal from the Oz channel. As presented in Figure 5, significant spectral responses appeared at the base frequency and its integer multiples across all ten experimental conditions, confirming that the visual stimulation paradigm used in this study effectively induced steady-state visual evoked potentials (SSVEPs) under various motion frequency conditions. The noticeable trend observed in the frequency domain analysis is the progressive attenuation of harmonic amplitudes as the harmonic order increases. In addition, compared with the static condition, the harmonic energy concentration during motion states was slightly weaker, and the amplitude at the fundamental frequency also showed a certain degree of attenuation. This phenomenon may be attributed to the discontinuous coverage of visual areas caused by stimulus movement, which introduces temporal disruptions in the perceived flicker sequence. Such disruptions can lead to Doppler-like distortions in the frequency spectrum, primarily manifested as a reduced fundamental frequency amplitude and dispersed harmonic energy. These results indicate that motion-related factors exert a measurable influence on the stability of frequency components during stimulus presentation. These findings are further supported by the signal-to-noise ratio (SNR) results shown in Appendix A.

Figure 6 highlights the notable discrepancies between the three motion trajectory conditions and the three motion speed conditions. Each factor was analyzed via one-way ANOVA based on a total sample size of 3672. A significant main effect was found for motion speed, F(2,3670)=8.121,p<0.001, as well as for motion trajectory, *F*(2, 3670) = 9.729, *p* < 0.001. Post hoc comparisons using Bonferroni correction revealed that, for motion trajectory, the square wave differed significantly (*p* < 0.05) from both the sinusoidal and sawtooth waves, while no significant difference was found between the latter two. For motion speed, the 1/20 speed condition differed significantly (*p* < 0.05) from both the 1/10 and 1/5 speeds.

Figure 7 shows the variation in decoding performance under different trajectory and speed combinations with increasing data length. In general, static trajectories consistently achieved the highest classification accuracy and ITR for the evaluated frequencies, demonstrating stable and superior performance. For example, for a 4 s time window, the static condition attained a 87.75 ± 13.78% average classification accuracy and a 20.82 ± 7.61 bits/min ITR, outperforming all other experimental conditions. In contrast, dynamic conditions exhibited a general decline in system performance, with accuracy decreasing as motion speed increased. Notably, among the various dynamic trajectories, the sinusoidal (sin20) and sawtooth (saw20) trajectories at low speed (1/20 of their flicker frequency) showed remarkable robustness. With a 4 s window, the sin20 trajectory achieved an average classification accuracy reaching 85.84 ± 13.96%, while the ITR reached 19.36 ± 7.69 bits/min—only about 3% lower than the static condition—indicating minimal interference and high signal quality. Similarly, saw20 reached 83.82 ± 11.11% accuracy and a 17.69 ± 6.18 bits/min ITR, also reflecting a relatively strong performance. In comparison, there was a clear performance disparity among different trajectories. The square wave trajectory (squ20) performed significantly worse under dynamic conditions. For the same 4 s window, squ20 yielded only 74.26 ± 15.81% accuracy and a 12.90 ± 6.98 bits/min ITR, substantially lower than both sin20 and saw20. This decline may be attributed to its irregular and abrupt motion pattern, which likely disrupted gaze stability and degraded the quality and recognizability of the SSVEP signals. In summary, the BCI system performs best under static conditions. However, in dynamic scenarios, trajectories characterized by smooth or periodic motion patterns—such as sin20 and saw20—can effectively maintain a high classification accuracy and ITR, suggesting lower levels of interference and greater practical potential. Based on the above findings, the static, sin20, and saw20 paradigms were selected as fundamental control commands in the subsequent online experiment.

Figure 8 depicts the impact of the various task types on subjects’ subjective cognitive load. Overall, there is a clear trend indicating that subjective ratings increased significantly with trajectory speed, peaking under high-speed conditions. This suggests that high-speed dynamic tasks substantially elevate subjects’ mental workload and perceived task difficulty.

From the perspective of trajectory type, compared to static tasks, certain dynamic trajectories—particularly low-speed sinusoidal (sin) and sawtooth (saw) patterns—exhibited lower subjective ratings. This indicates that such trajectories may help alleviate cognitive load and psychological stress to some extent. In contrast, square wave (square) tasks consistently yielded higher subjective scores than static tasks across all speed levels, suggesting a heavier cognitive burden.

Further support for these findings is provided by the heatmap analysis. Under low-speed conditions, tasks with sinusoidal and sawtooth trajectories show statistically significant differences (*p* < 0.05) in subjective ratings compared to static tasks, reinforcing their potential in reducing user load. However, as speed increases—especially under high-frequency conditions and across all square wave tasks—the differences in scores between task types become more pronounced. This highlights a notable interaction between trajectory speed and type in influencing cognitive load.

In summary, low-frequency sinusoidal and sawtooth tasks demonstrate promising potential for reducing subjects’ subjective workload, making them suitable for low-stress brain–computer interface (BCI) applications. Conversely, high-speed dynamic tasks—particularly those involving square wave stimulation—may impose a considerable psychological burden and should be applied with caution or in combination with other optimization strategies to enhance user experience.

### 3.3. Online Experiment Results

Based on the above offline analysis, an online SSVEP-based fruit-slicing game was developed using three conditions—static, sin20, and saw20—with each trial lasting 4.5 s. Table 2 illustrates the BCI performance of 12 subjects during the online experiment. Under the static condition, the average classification accuracy across all subjects was 84.17 ± 13.11%. Similarly, the average accuracies for the sin20 and saw20 conditions were 82.92 ± 9.88% and 84.17 ± 10.84%, respectively, showing performance comparable to the static condition. This comparison highlights the robustness of the periodic motion patterns in the sin20 and saw20 conditions, which effectively maintain system performance in dynamic scenarios. These findings demonstrate the feasibility of the proposed dynamic SSVEP stimulation encoding method that simultaneously modulates both trajectory and speed.

Figure 9 presents SUS scores, with the left panel illustrating the mean scores and standard deviations for individual questionnaire items, and the right panel displaying the composite SUS score (maximum 100) calculated using standardized protocols. Positively worded items (Q1, Q3, Q5, Q7, Q9) demonstrated strong performance, achieving mean scores above 3.8/5 (Q1: 4.23 ± 0.75; Q3: 4.15 ± 0.89), reflecting high user satisfaction with the system’s usability. Negatively phrased items (Q2, Q4, Q6, Q8, Q10) exhibited systematically lower scores, notably Q10 (“The system was very cumbersome”) with the lowest mean score (1.46 ± 0.52), where 92.3% of subjects strongly disagreed, further validating the system’s user-friendly design. The global SUS score reached 75.96 ± 7.88, exceeding the benchmark threshold of 68 for acceptable usability and categorizing the system within the “Good” range (70–85.4), nearing the “Excellent” classification (>85.5). This outcome highlights the system’s strengths in learnability, operational efficiency, and user satisfaction, as evidenced by the high scores in critical metrics such as reverse-scored Q4 (4.08 ± 0.86 for learnability) and Q7 (4.31 ± 0.63 for satisfaction). The coherent alignment between elevated positive item ratings and suppressed negative item scores underscores the system’s successful integration of technical robustness and human-centered design principles, positioning it as a viable solution for practical SSVEP-BCI applications requiring sustained user engagement.

## 4. Discussion

This study comprehensively investigated the impact of dynamic trajectory patterns and speed modulation on both performance metrics and subjective user experience in SSVEP-BCI systems while validating their feasibility and practicality through an online fruit-slicing game integrated with dynamic stimulation paradigms. Experimental results demonstrated that, although conventional static stimuli maintained superior performance in core technical indices—including evoked signal intensity, classification accuracy, and Information Transfer Rate (ITR)—dynamic paradigms under optimized motion parameters (notably, sinusoidal and sawtooth waves modulated at 1/20 of the flicker frequency) achieved near-static performance levels while significantly enhancing user comfort and operational experience. These findings establish a theoretical foundation and practical framework for developing novel SSVEP-BCI systems that harmonize neural decoding efficiency with human-centered interaction design, particularly demonstrating the viability of trajectory–speed co-modulation strategies in balancing neurophysiological signal stability and ergonomic optimization for real-world applications requiring sustained attentional engagement.

In contrast to previous studies predominantly employing linear unidirectional motion or random movement paradigms, this research pioneers a systematic investigation into the mechanistic impacts of periodic trajectories (sinusoidal, sawtooth, and square waves) under varying speed modulations on SSVEP characteristics. Offline experimental outcomes revealed that static conditions achieved superior flicker frequency amplitudes (4.2 µV at 6 Hz), classification accuracy (87 ± 13% with 4 s windows), and ITR (20 ± 7 bits/min) across all target frequencies, aligning with Li et al.’s conclusion regarding the performance supremacy of static paradigms under dual luminance–motion modulation. Crucially, low-speed dynamic stimuli—specifically sinusoidal (sin20) and sawtooth (saw20) waves—exhibited negligible performance deviations from static conditions (85 ± 13% accuracy in 4 s windows with <3% difference from static, ITR variance of 1–3 bits/min) while significantly outperforming static stimuli in subjective evaluations. These findings demonstrate strong consistency with Duan et al.’s [22] proposition that “low-frequency motion paradigms can achieve performance parity with static conditions,” while further extending the theoretical framework by quantitatively establishing the neuroergonomic advantages of periodic motion patterns in balancing technical performance and user comfort for human-centric BCI applications.

At the level of motion trajectories, low-speed and smooth patterns—such as the sinusoidal wave at 1/20 frequency (sin20)—may enhance SSVEP signals by introducing rhythmic and dynamic visual changes that help attract and sustain subjects’ attention. This finding echoes the perspective of Duan et al. [22], who suggested that dynamic stimuli can enhance the persistence of visual perception. Moreover, significant performance differences were observed across different trajectory shapes. Under low-speed modulation (1/20), sinusoidal and sawtooth waveforms yielded the highest classification accuracies (85.84 ± 13.96% and 83.82 ± 11.11%, respectively). In contrast, square waveforms consistently demonstrated the lowest performance across all speed conditions—for instance, achieving only 74.26 ± 15.81% accuracy in a 4 s analysis window. These results further suggest that the continuity and smoothness of motion trajectories are critical factors influencing the stability of SSVEP signals and the efficiency of their elicitation.

In terms of user experience, this study employed the NASA-TLX workload assessment scale for subjective evaluation. The results revealed that low-speed dynamic stimulation conditions (sin20 and saw20) significantly outperformed static, high-frequency, and non-smooth motion paradigms across key metrics including cognitive load, visual fatigue, and perceived task redundancy, highlighting the potential of dynamic paradigms in enhancing human–computer interaction comfort. These findings align with Punsawad et al.’s proposition that “dynamic stimuli alleviate visual fatigue and improve user experience,” emphasizing the critical value of dynamic stimulation design within human-centered interaction frameworks. The observed neuroergonomic advantages—particularly the dissociation between near-static-level technical performance and superior subjective ratings—provide empirical evidence for prioritizing periodic motion parameters in BCI applications requiring sustained user engagement, thereby advancing the integration of psychophysiological optimization principles into interactive system design.

The online experiment further validated the applicability of the dynamic paradigm in practice. A fruit-slicing game system was developed based on sin20 and saw20, achieving average classification accuracies of 83.84 ± 12.60% and 84.17 ± 10.84%, respectively, as shown in Figure 10, without requiring additional training or parameter adjustment. These results showed no significant difference compared to the static condition. Additionally, to assess the feasibility of the proposed paradigms, an online experimental setup from Li [24] was adopted as a benchmark, introducing a horizontal movement condition at 0.2 Hz. As illustrated Figure 10, under 0.2 Hz, the classification accuracy dropped significantly to 76.25 ± 9.66%, representing a decrease of 7.59 and 7.92 percentage points compared to the sin20 and saw20 conditions, respectively.

Furthermore, after the experimental session, subjects filled out a questionnaire corresponding to each stimulation condition. The questionnaire consisted of three dimensions [34]:1.A comfort scale scored on a 1–5 scale (1–5: very uncomfortable–very comfortable);2.A flicker perception scale scored on a 1–5 scale (1–5: very annoying–imperceptible);3.A preference scale scored on a 1–5 scale (1–5: very disgusting–very likeable).

Paired-sample *t*-tests were conducted to analyze the questionnaire scores obtained under the three experimental conditions. As shown in Table 3, compared to 0.2 Hz, both sin20 and saw20 exhibited higher scores across the three subjective rating dimensions: comfort, flicker perception, and preference. The majority of these results were statistically significant. Specifically, sin20 demonstrated significantly higher comfort scores (3.54 ± 0.88) than 0.2 Hz (*p* < 0.001), and even more pronounced differences were observed in flicker sensation and preference, with both reaching highly significant levels (*p* < 0.001), with scores of 3.77 ± 0.83 and 3.77 ± 0.60, respectively. saw20 performed even more impressively, achieving the highest scores across all three rating dimensions, particularly in comfort, where it reached 3.92 ± 1.12, which was also significantly higher than 0.2 Hz (*p* < 0.01). It can be inferred from the results that, compared to the dynamic paradigm designed by Li, the proposed design in this study offers a superior enhancement to the users’ subjective experience, potentially mitigating the effects of visual fatigue or discomfort commonly associated with SSVEP paradigms. Overall, this outcome not only highlights the robustness of the dynamic paradigm in complex task environments but also reflects its superior user adaptability in practical applications.

The achievement of these results can be primarily attributed to three key design optimizations: First, the natural integration of stimulus trajectories with interactive game elements significantly reduced the cognitive load associated with attention shifting and target recognition. Second, by aligning motion speed with target frequency, the design effectively avoided potential intermodulation interference. Third, the incorporation of dynamic visual feedback within the game—such as the fruit-slicing animation—reinforced the neurofeedback loop between motion and perception, potentially enhancing both the consistency of SSVEP elicitation and neural plasticity. Collectively, these design elements contributed to a System Usability Scale (SUS) score of 75.96, which is notably higher than the average benchmark of 68, thereby validating the proposed dynamic paradigm’s practical advantages and innovative value in enhancing the BCI experience.

Overall, this study presents several notable innovations that advance the field of dynamic SSVEP-based BCI research: First, it is the first to systematically explore the combined effects of various periodic trajectory patterns and speed modulation strategies on SSVEP elicitation and BCI performance, significantly broadening the design parameter space for dynamic SSVEP paradigms and offering new insights into optimizing stimulus configurations. Second, by deeply integrating dynamic visual stimuli with concrete task scenarios—such as interactive gaming—and embedding a real-time feedback mechanism, the study achieves a closed-loop system that bridges signal induction and user perception. This marks a substantial departure from conventional approaches where dynamic stimuli are merely presented as passive control signals, thereby enhancing both the functionality and ecological validity of BCI applications. Third, the research empirically validates the adaptability and feasibility of the joint trajectory–speed modulation strategy in dynamic visual environments, especially in contexts involving spatial uncertainty or requiring active user interaction. The demonstrated performance in such complex settings provides not only strong evidence for the paradigm’s robustness and practicality, but also a strong theoretical underpinning for next-generation BCI systems that prioritize user engagement, intuitive control, and immersive experience.

Despite the promising progress achieved in this study, several limitations remain and warrant further investigation in future work. First, the sample sizes in the offline and online experiments were relatively limited, with 17 and 12 subjects, respectively, and some overlap between participants. This may limit the generalizability of the findings to a broader population. Due to constraints in experimental design and available resources, we conducted an initial exploratory study with the current sample. Future research will aim to expand the sample size to include a wider age range and clinical populations, thereby improving the representativeness and applicability of the results. In addition, although ANOVA was used to compare different trajectory and speed conditions, Bonferroni post hoc tests were specifically applied to control for Type I error due to multiple comparisons, ensuring the statistical reliability of the results. Nevertheless, the limitations of the statistical methods should be acknowledged—especially given the multi-condition comparisons involved in dynamic SSVEP paradigms. Future work may consider adopting more advanced multiple comparison correction techniques and statistical models to further validate the robustness of the findings. Overall, despite the above limitations, this study provides valuable parametric benchmarks and methodological references for future research on dynamic stimulation modulation and real-time BCI applications and holds meaningful exploratory value and application potential. In the future, we will focus on expanding participant diversity and employing more comprehensive statistical approaches to enhance the robustness of our conclusions.

Second, this study employed three typical periodic motion trajectories (sinusoidal, square, and sawtooth waves) and three linear speed ratios (1/5, 1/10, and 1/20 of the flicker frequency) to construct a relatively systematic and representative parameter space for dynamic stimulation. This design facilitated a structured evaluation of how trajectory and speed influence SSVEP performance. However, the study did not include non-periodic trajectories (elliptical or spiral paths) or nonlinear speed profiles (gradual acceleration or abrupt deceleration), which limits the ecological validity of the findings in complex natural visual motion environments. The selection of the above parameters was primarily based on technical and theoretical considerations—specifically, ensuring experimental controllability, repeatability, and analytical stability. We believe that establishing such a basic and well-defined parameter framework is a critical first step in the design of dynamic SSVEP paradigms, providing a valuable foundation for future exploration of more complex motion patterns and speed variations. Future studies will build upon this framework to expand toward more diverse motion trajectories (non-periodic paths) and nonlinear speed modulations, better approximating real-world visual dynamics.

Third, this study did not include real-time monitoring or control of physiological artifacts that may affect SSVEP signal quality, such as eye movements, head motion, and muscle activity noise. This limitation was primarily due to the constraints of the experimental setup, which lacked equipment such as eye trackers, electrooculography (EOG), or surface electromyography (sEMG) devices. To minimize interference, we applied filtering techniques during data preprocessing to partially suppress these artifacts. In future research, we plan to incorporate eye tracking, EOG, and sEMG monitoring to enable real-time detection and control of physiological artifacts, thereby further improving data quality and the reliability of the results.

Fourth, a fruit-slicing game was used as the application scenario in the online validation phase of this study. Although the task involved relatively low interaction complexity, it was deliberately chosen to create a controlled, low-noise environment that allowed us to focus on the fundamental mechanisms of trajectory and speed modulation in dynamic SSVEP-based BCI control. This ensured that the evaluation of system responsiveness and real-time closed-loop performance was not confounded by excessive external variables. Due to limitations in experimental resources and hardware compatibility, we were not able to implement more complex and ecologically valid interaction paradigms, such as virtual reality navigation or multimodal attention-shifting tasks. Nevertheless, the simplified task structure enabled a clear and stable analysis of system performance. More importantly, this design provides a solid foundation for future extensions. Future research will build upon this platform by progressively introducing more complex and ecologically relevant interaction tasks—such as virtual reality navigation, multi-target attention switching, or multimodal control—to further validate the robustness and practicality of the dynamic SSVEP paradigm in real-world applications.

Fifth, this study primarily employed the NASA-TLX and SUS to assess users’ workload and system usability. Both instruments are internationally recognized standardized tools with high reliability and validity. Their concise structure allows for quick and easy completion, making them particularly suitable for experiments such as brain–computer interface (BCI) studies, where participants’ cognitive load is sensitive. Moreover, the widespread application of the NASA-TLX and SUS in related fields helps improve the comparability and reproducibility of research results. However, these standardized tools also have certain limitations. First, the NASA-TLX and the SUS focus on overall workload and usability but do not cover more subtle and important subjective dimensions such as perception of stimulus rhythm, aesthetic quality of trajectories, accumulation of visual fatigue, and long-term usability. Especially in the context of this study’s emphasis on user-centered design, these unassessed factors may have significant impacts on user experience and system acceptance. Second, although there are tools such as the User Experience Questionnaire (UEQ) and Visual Aesthetics of Websites Inventory (VisAWI) that can more comprehensively capture users’ emotional and aesthetic responses, their longer questionnaire formats and higher cognitive burden are not suitable for the short-duration, multi-trial experimental procedures used in this study. Qualitative methods such as semi-structured interviews can deeply explore user preferences but lack quantitative standards and increase experimental complexity. Considering experimental time constraints and participant burden, this study balanced assessment comprehensiveness and experimental feasibility by selecting the NASA-TLX and SUS as the subjective evaluation tools. Nonetheless, we explicitly acknowledge in Section 4 that the dimensions not covered by these tools constitute a limitation of this study. Future research plans include the introduction of more detailed questionnaires, semi-structured interviews, and physiological measurements to systematically investigate visual fatigue, rhythm perception, and long-term usability, thereby enhancing the depth and breadth of user experience evaluation.

Sixth, in addition to the gaming and human–computer interaction applications demonstrated in this study, the proposed dynamic SSVEP paradigm holds considerable potential for various clinical scenarios. For example, attention training for individuals with attention deficit hyperactivity disorder (ADHD) could benefit from low-fatigue and engaging visual stimuli to improve sustained focus. Similarly, cognitive intervention programs for older adults aimed at delaying cognitive decline may find the paradigm useful for maintaining motivation and adherence. Moreover, neurorehabilitation following stroke or brain injury often requires repetitive, cognitively demanding tasks; the dynamic and customizable features of our paradigm could reduce user fatigue and increase training effectiveness. Future research should actively explore these clinical applications, incorporating longitudinal studies with patient populations and multidisciplinary collaborations. By doing so, the paradigm could not only advance BCI technology but also contribute meaningfully to rehabilitation and cognitive health, thereby expanding its societal relevance and impact.

In summary, these findings mark important progress in designing dynamic SSVEP stimuli, improving user experience, and developing practical interaction frameworks. By enriching the conceptual landscape of SSVEP-based BCI systems and offering robust theoretical and experimental foundations, it paves the way for the development of future research characterized by higher adaptability, richer interactivity, and broader applicability across both general and clinical user populations.

## 5. Conclusions

This study proposes a dynamic SSVEP paradigm depending on the joint modulation of trajectory and speed, systematically investigating the impact of periodic motion parameters on both system efficiency and user satisfaction. The findings demonstrate that this paradigm achieved performance comparable to that of traditional static stimuli while significantly reducing users’ cognitive workload, as evidenced by lower NASA-TLX scores, thereby striking a favorable balance between system efficiency and user comfort. Moreover, sinusoidal and sawtooth trajectories, owing to their greater motion continuity and lower signal interference, significantly outperformed square wave trajectories, underscoring how crucial trajectory design is for maintaining the stability of SSVEP responses. In the online experiment, the fruit-slicing system based on dynamic stimuli achieved real-time classification accuracies of 82.92 ± 9.88% (sinusoidal) and 84.17 ± 10.84% (sawtooth), along with a System Usability Scale (SUS) score of 75.96—well above the standard benchmark of 68—highlighting its strong usability. Collectively, the results underscore the practical application potential of the dynamic paradigm and provide a solid theoretical foundation for the parameter optimization and user-centered enhancement of dynamic SSVEP-based BCI systems.

## Figures and Tables

**Figure 1 sensors-25-04727-f001:**
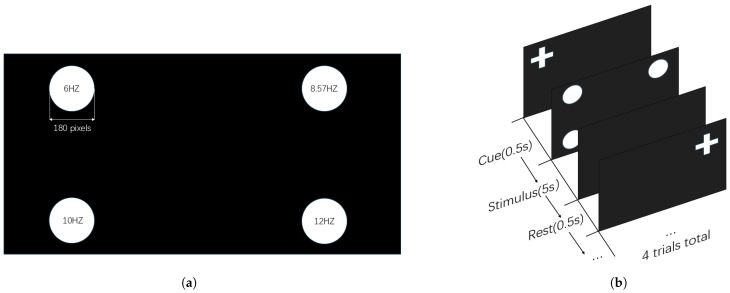
(**a**) Offline experimental design. (**b**) The sequence of SSVEP stimulation trials presented within a single experimental block.

**Figure 2 sensors-25-04727-f002:**
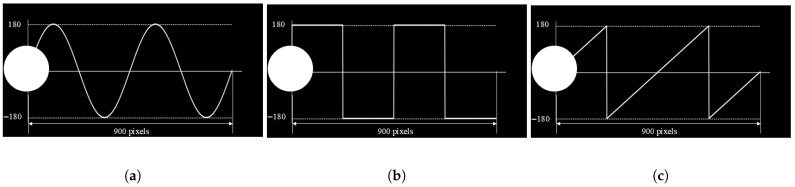
(**a**) Sinusoidal wave. (**b**) Square wave. (**c**) Sawtooth wave.

**Figure 3 sensors-25-04727-f003:**
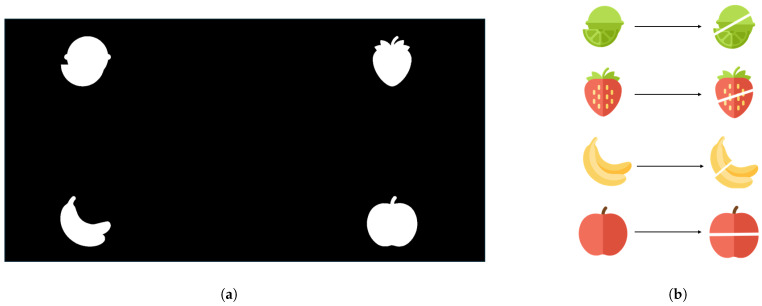
(**a**) Online experimental design. (**b**) The slicing feedback presented upon successful recognition of each type of fruit.

**Figure 4 sensors-25-04727-f004:**
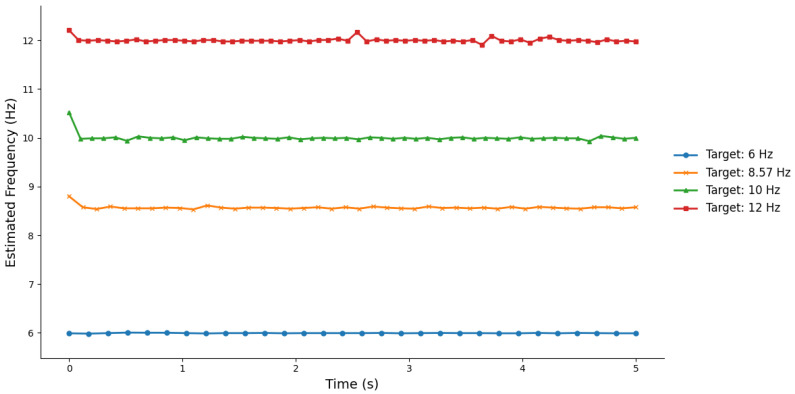
Frequency fluctuations of four flicker frequencies within 5 s.

**Figure 5 sensors-25-04727-f005:**
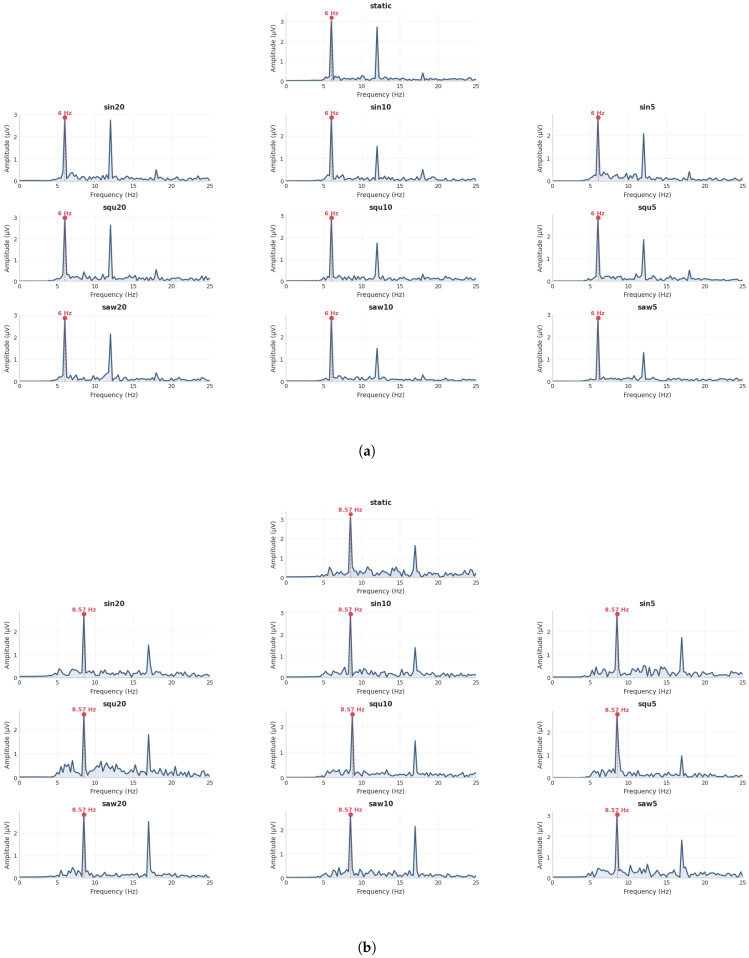
SSVEP responses are represented by the average amplitude spectra derived from channel Oz under different combinations of trajectory and speed. Panels (**a**–**d**) depict the SSVEP amplitudes corresponding to the four flickering frequencies, respectively.

**Figure 6 sensors-25-04727-f006:**
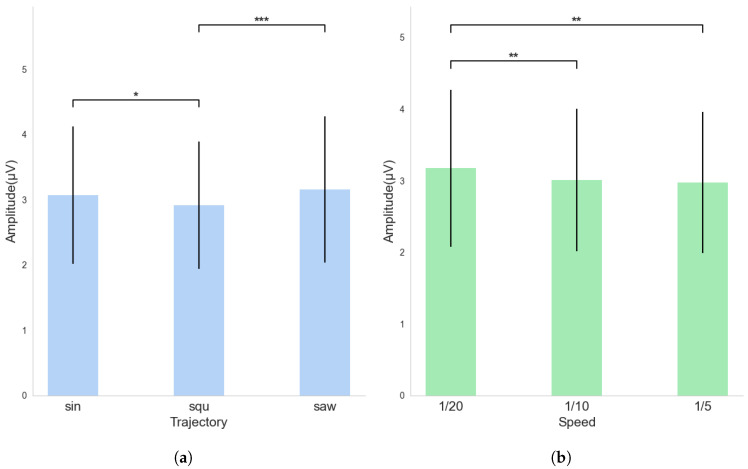
(**a**) Average SSVEP amplitude values for each trajectory condition. (**b**) Average SSVEP amplitude values for each speed condition (* *p* < 0.05, ** *p* < 0.01, *** *p* < 0.001).

**Figure 7 sensors-25-04727-f007:**
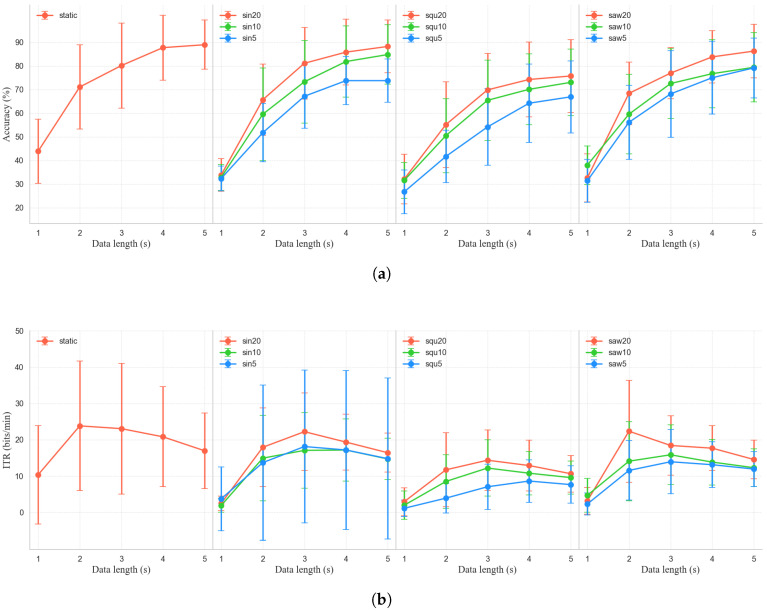
(**a**) Mean recognition accuracy and (**b**) ITR across different trajectory and speed settings with various data lengths.

**Figure 8 sensors-25-04727-f008:**
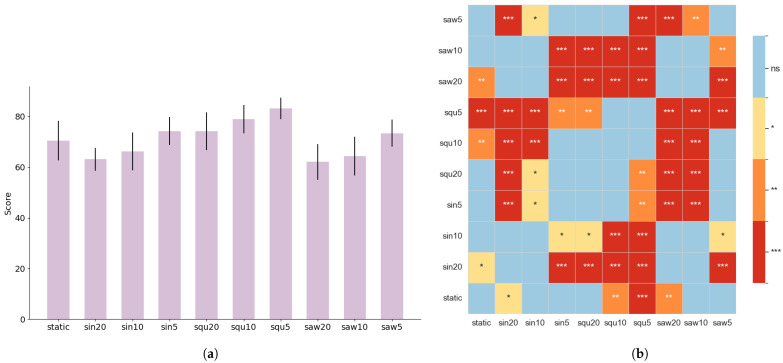
(**a**) Average NASA-TLX scores under different trajectory and speed conditions. (**b**) Heatmap of NASA-TLX scores across various speed and trajectory conditions. Color represents statistical significance: * *p* < 0.05, ** *p* < 0.01, *** *p* < 0.001.

**Figure 9 sensors-25-04727-f009:**
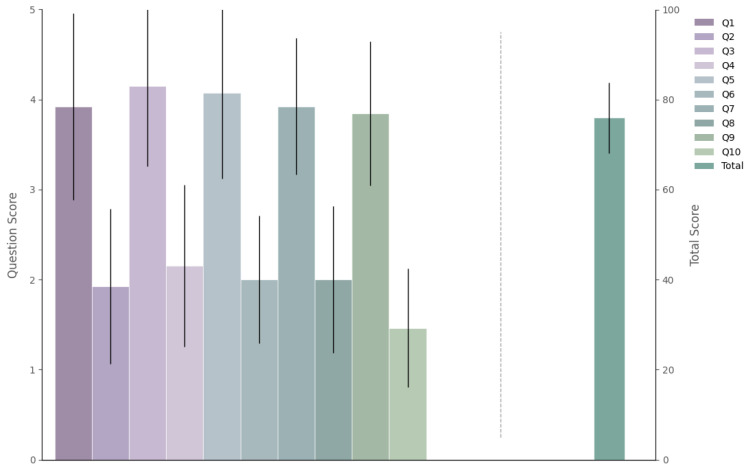
SUS scores from 12 subjects. The left panel shows the average score for each question, while the right panel presents the average overall score.

**Figure 10 sensors-25-04727-f010:**
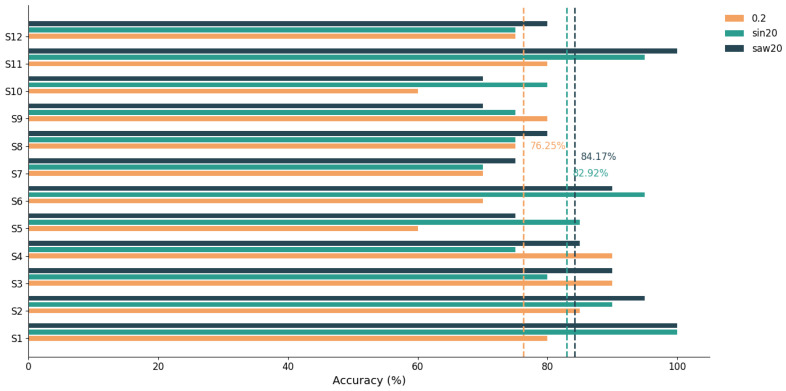
The accuracy rates of 12 subjects under 0.2 Hz, sin20, and saw20, with the dashed line representing the average accuracy.

**Table 1 sensors-25-04727-t001:** Summary of flicker frequency verification results.

Flicker Frequency (Hz)	Test Duration (s)	N	Average Frequency (Hz)
6.00	5	30	5.9951 ± 0.0043
8.57	5	42	8.5700 ± 0.0398
10.00	5	50	10.0023 ± 0.0773
12.00	5	60	12.0013 ± 0.0432

**Table 2 sensors-25-04727-t002:** Online recognition accuracy for 12 subjects. Subjects with a * engaged in both experiments.

Subjects	Accuary(%)		
	**static**	**sin20**	**saw20**
S1 *	90	100	100
S2	95	90	95
S3 ^*^	65	80	90
S4	90	75	85
S5	100	86	75
S6	85	95	90
S7 *	60	70	75
S8 *	75	75	80
S9	80	75	70
S10	80	80	70
S11	100	95	100
S12	100	75	80
Average ± SD	84.17 ± 13.11	82.92 ± 9.88	84.17 ± 10.84

**Table 3 sensors-25-04727-t003:** Questionnaire scores under the three conditions (* *p* < 0.05, ** *p* < 0.01 ).

	Score		
	**Comfort**	**Flicker Perception**	**Preference**
0.2	3.08 ± 0.95	3.07 ± 0.64	3 ± 0.58
sin20	3.54 ± 0.88 **	3.77 ± 0.83 **	3.77 ± 0.60 **
saw20	3.92 ± 1.12 *	3.85 ± 0.90 *	3.77 ± 0.60 **

## Data Availability

The data presented in this study are available on request from the corresponding author (the data are not publicly available due to privacy or ethical restrictions).

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
