# Peer review of "Optimization of Dynamic SSVEP Paradigms for Practical Application: Low-Fatigue Design with Coordinated Trajectory and Speed Modulation and Gaming Validation"

_sensors, 2025, doi:10.3390/s25154727_

Round 1
Reviewer 1 Report
Comments and Suggestions for Authors
In their Manuscript „Optimization of Dynamic SSVEP Paradigms for Practical Application: Low-Fatigue Design with Coordinated Trajectory and Speed Modulation and Gaming Validation“ the authors propose and evaluate the use of SSVEP stimuli which are side shifted while flickering. The authors propose to use a sinusoidal or a triangular left to right to left movement. They test different frequencies at which the stimuli move. The achieved accuracies of about 80 $ in the offline and online experiment are quite reasonable but fall behind typical accuracies of > 90 % achieved with standard SSVEP BCI systems.
Apart from the promising approach to allow fast long term use of SSVEP BCI’s the following issues have to be resolved before the manuscript can be published!
1) page 1 line 20 Introduction
“[…] driving innovation in human-computer interaction […]”
I do doubt the claim that BCI systems are driving the innovation in Human-Computer interaction. Given the efforts to setup, and operate, their main use cases are
-
Rehabilitation of neuro degenerative diseases
-
assistive system
-
Marketing, User Acceptance-Engagement studies to quantify and verify the results of qualitative assessments
In any other cases where Humans still have full or at least sufficiently complete control of their body any other means of Human computer interaction is more direct, reliable and requires a signifiant less amount of efforts compared to BCI based interaction.
So please either elaborate and clarify this claim or drop.
2) Page 1 Line 30
“[…] static visual stimuli […]”
Apart from the typo (visiual), I do guess with static visual stimuli the authors mean stimuli with a fixed steady position on the screen. As static could also mean not flickering please either reword or or be elaborate on what is meant by static in this context.
3) Page 3 Lin 85
“[…] four-frequency stimulation (6 Hz, 8.57 Hz, 10 Hz, 12 Hz) , with three speed
modes (1/20, 1/10, and 1/5 of the fundamental frequency) […]”
I do guess the 6 Hz is a typo here and should at least read 6.5Hz or better 6.67Hz. If the authors really would have used 6Hz and 12 Hz, which is the first harmonic of 6 Hz, I would expect accuracies a lot less 80 %. So please correct and check that naming of frequencies is consistent throughout the manuscript.
Compare page 3 line 129
4) Page 3 Lines 93-104
These lines are a 1:1 duplicate of the preceding lines 81-92 (page 2 – page 3) remove them.
Page 3 Materials and Methods line 109
“[…] he trial protocols were developed using web technologies including HTML, JavaScript, and CSS […]”
I do doubt that Html, javascript and css are precise enough to deliver proper stimuli, with sufficiently narrow frequency bands. Especially it is to be doubted that the refresh rate of the browser is sufficiently in sync with screen refresh. So please also provide results on stimulus accuracy, frequency spectrum of each stimulus and frequency variation. This is necessary as the frequencies selected by the Authors are quite close to each other with a minimum gap of less than 2 Hz.
In the current form it is to be questioned that the stimuli flicker at the proposed frequencies at all. A broader frequency spectrum or a centre frequency which deviates too much from the intended one would explain why accuracies are as low as 80 %.
5) Page 4 Equation 1
Why using frequencies 6.5 Hz, 8.57 Hz, 10 Hz, 12 Hz when anyway using brightness modulation to simulate continuous sinusoidal stimulus. If synchronization as stated under 4) would be exact enough than any combinations of non harmonic frequencies could be used for the stimuli. I would at least use 15, 12 , 10, 6.67 Hz to ensure sufficiently enough distances between stimuli.
6) Page 4 line 145:
Please explain what the fundamental frequency is. Is it the flicker frequency of the stimulus or the screen refresh rate ?
7) As the stimuli are moving how many pixels they are shifted left to right right to left?
8) Dependent upon how much the stimuli are shifted please also discuss how far move of stimuli elicits in pixels not covered full time by stimulus Doppler like effects and their impact upon SSVEP spectrum of stimulus, its harmonics and detection accuracy.
9) Page 8 Figure 3.
Please convert the 3D plots into normal 2D plots. In their current form they are not properly interpretable and thus use less to the reader.
10) Why is only the Oz Channel used for analysis instead of using all channels individually or merging them to improve detection accuracy and robustness? How would Sample based merging or temporal chaining of channels improve or worsen the achieved results? Canonical Correlation Analyses can be trained to analyse all channels simultaneously as done for phase shifted SSVEP Stimuli.
Comments on the Quality of English LanguagePlease proofread and correct typos.
Reviewer 2 Report
Comments and Suggestions for Authors
In this study, Huang et al. explored the parameter space of motion modulation in SSVEP applications. They identified a set of parameters, i.e. slow sine and sawtooth motions, that reduced cognitive loads without performance loss when compared to stationary stimulation. Though the size of reduction is relatively minor (~10%), it is useful for guiding future SSVEP BCI designs and the overall exploration and characterization can be informative.
I do not have major concerns with this study. My comments below are mostly about clarification.
Since this study involves two types of periodic modulations, it would be helpful to use clear qualifications in terminologies. Also make sure the same terminologies are used consistently throughout the paper, avoid unnecessary interchangeable uses.
- Consider changing "stimulation frequency" to something more descriptive and specific such as "flicker frequency", "luminance frequency", or something similar.
- Consider replacing "fundamental frequency" with the above term of choice, or clearly define the above term as "base frequency".
Are motion trajectories on the screen 1D or 2D? Seems like this is 1D, but is it horizontal or vertical? What is the range of motion on the screen for each stimulus? Please clearly illustrate like, for instance, Duan et al. 2021.
Consider making a ladder of 2D plots (x-axis freq, y-axis amp) for each of the panels in Figure 3. The 3D plots are hard to read and unnecessary. To better show how parameters in each category modulate the spectrum, consider plot static on the first row of the ladder, an overlay of three sin (with different colors) on second row, three squ on third, three saw on fourth.
Line 249-251: Again, it is almost impossible to see what the authors are describing from the small and skewed 3D plots.
Figure 4. ANOVA tests the existence of difference across multiple samples but does not test differences pairwise. Please explain how pairwise differences were tested here. If the authors applied ANOVA on two samples at a time, consider using two-sampled test options directly. Also please report the sample size (N) of each sample and other test information in figure legend.
Similar issues regarding ANOVA use apply to a later analysis in Table 2.
The relationship between motion parameters and operatability (SUS) is unclear.
Reviewer 3 Report
Comments and Suggestions for Authors
Comments attached.

Round 2
Reviewer 1 Report
Comments and Suggestions for Authors
In their Manuscript „Optimization of Dynamic SSVEP Paradigms for Practical Application: Low-Fatigue Design with Coordinated Trajectory and Speed Modulation and Gaming Validation“ the authors propose and evaluate the use of SSVEP stimuli which are side shifted while flickering. The authors propose to use a sinusoidal or a triangular left to right to left movement. They test different frequencies at which the stimuli move. The achieved accuracies of about 80 % falling behind typical accuracies of > 90 % achieved with standard SSVEP BCI systems my be explained by the significantly shorter evaluation period and additional effects introduced by motion.
The quality of the manuscript has been significantly improved by the authors, never the less some questions and remarks remain to be addressed and solved before it can be published.
1) In the introduction the authors still list 6.5 Hz as lowest stimulation frequency instead of 6 Hz as confirmed by the Authors in their response to the first report.
2) I still do question that the use of 6Hz and its harmonic of 12 Hz unnecessarily makes stimulus detection unnecessarily more challenging and difficult. And is one if not the mayor cause why classification results are as low as 80% instead of close to 90 % and thus comparable with standard SSVEP systems using evaluation periods of 2 seconds and more.
3) figure 1b please add arrow head to time line clearly indicating direction of paradigm progress. This would make it easier to read and understand the diagram.
4) Figure 1b) I do guess that each trial consisting of cut, stimulus rest phase is repeated 4 times thus please fix typo "4 trails total" and change to "4 trials total".
5) The description of the motion trajectory is not plausible. Especially a motion range of 900 pixels for each stimulus is quite large given a screen of 2560 pixels in width and 1440 pixels in height. that would result in
- ( 900 - 180 )/2 = +-360 pixels
More plausible and realistic is that the amplitude shown in figure 2 on the y axis , is the number of pixels the stimuli are moved off their central position and that the 900 pixels shown on the x axis are actually the number of frames (screen frames) the motion trajectory is computed for before it is replayed from the start.
given a screen refresh-rate of 60Hz the 900 frames would represent 15 seconds meaning one motion cycle would cover 7,5 seconds and thus result in a frequency of 0,13 Hz which still would b a bit low even when flicker to motion frequency would be 1/20th so possibly its 90 frames instead of 900 frames.
Please cross check with your implementation and update accordingly. In the current form it is not plausible and difficult to reproduce.
